# Optimizing Network Resilience via Vertex Anchoring

## ABSTRACT

Network resilience is a critical ability of a network to maintain its functionality against disturbances. A network is resilient/robust when a large portion of the nodes are to be better engaged in the network, i.e., they are less likely to leave given the changes on the network. Existing studies validate that the engagement of a node can be well captured by its coreness on network topology. Therefore, it is promising to maximize the number of nodes with increasing coreness values. In this paper, we propose and study the *follower maximization* problem: maximizing the resilience gain (the number of coreness-increased vertices) via anchoring a set of vertices within a given budget. We prove that the problem is NP-hard and W[2]-hard, and it is NP-hard to approximate within an $O(n^{1-\epsilon})$ factor. We first propose an advanced greedy approach, followed by a time-dependent framework designed to quickly find high-quality results. The framework is initialized by the advanced greedy algorithm and incorporates novel techniques for optimizing the search space. The effectiveness and efficiency of our solution are verified with extensive experiments on 8 real-life datasets.

**ACM Reference Format:**
Anonymous Author(s). 2024. Optimizing Network Resilience via Vertex Anchoring. In *Proceedings of the ACM Web Conference 2024 (WWW '24)*. ACM, New York, NY, USA, 13 pages. https://doi.org/XXXXXXX.XXXXXXX

## 1 INTRODUCTION

Network resilience refers to a network's ability to adapt and endure changes, where node/user engagement is a key issue [21]. Many real-life networks are susceptible to dynamics [20], e.g., in a social network, there are often natural failures (the random leave of users due to their individual situations) and artificial attacks on the network (user attraction strategies from competing networks). The departure of users may contagiously affect the engagement of other users [38], which may even lead to the collapse of a network [21, 47]. Correspondingly, the users with increasing engagement may encourage the participation of other users who are thus less likely to leave the network. A network is resilient/robust if few nodes will leave the network given the negative changes. Thus, in order to sustain the resilience of a network, it is critical to identify and enhance node engagement to the greatest extent, e.g., maximize the number of engagement-enhanced nodes.

Real-world networks are usually modeled as graphs in which different graph characteristics are considered for the resilience study, e.g., centrality, connectivity, and diameter [20]. Despite the various

metrics proposed in the literature, as we will discuss in Section 2, most of the metrics do not consider the engagement dynamic of nodes which is closely correlated with network resilience.

The $k$-core is a widely studied cohesive subgraph model for user engagement analysis, which is defined as a maximal subgraph where each vertex has at least $k$ neighbors [39, 45]. Every vertex in the graph has a unique *coreness* value, i.e., the largest $k$ such that the $k$-core contains the vertex. The *core decomposition* can compute the coreness of every vertex by iteratively removing the vertices with the smallest degree in the remaining graph. This procedure well captures the engagement dynamic of users in the unraveling of a network, and thus the coreness is validated as the "best practice" for measuring user engagement on graph structure [38]. As shown in Figure 1, in Gowalla social network [33], there is a clear correlation between coreness $c$ and node engagement (represented by the average number of check-ins for all nodes with coreness $c$). Note that the correlation is also validated in other networks [50] and the outliers are due to the sparsity of the vertices with the same coreness.

Vertex anchoring is a common practice in recent studies to optimize the engagement of targeted users and improve the engagement of other users through contagious user interactions [6, 10, 16, 35, 37, 54]. We may provide incentives to key users s.t. they will be first "anchored" regarding user engagement and thus enhance the overall engagement of all the users. The degree of each anchored vertex can be considered as positive infinity, while its connections to other vertices are not changed, i.e., the anchored vertex will not be deleted in any batch of core decomposition. The coreness values of non-anchored vertices may be increased by vertex anchoring which reflects the true dynamic of user engagement.

In the literature, different objectives are proposed to optimize overall user engagement by anchoring a number of vertices, e.g., Bhawalkar et al. [6] propose the anchored $k$-core problem to maximize the size of $k$-core for a given $k$ value; Linghu et al. [35] study the anchored coreness problem to maximize the overall *coreness gain* of all the non-anchored vertices. However, the target of the above studies is different from the network resilience optimization studied in this paper. The natural failures or the attacks on a network may incur in a "random" manner, e.g., the collapse of the Friendster network may start from the leaving of users with either large corenesses [46] or relatively small corenesses [21]. Therefore, in order to optimize network resilience, we need to enhance the engagement of as many users as possible.

In this paper, we pursue the coverage of followers, i.e., the vertices with coreness increased in core decomposition with anchored vertices. We propose and study the *follower maximization* (FM) problem: given a graph $G$ and a budget $b$, anchor a set of at most $b$ vertices in $G$ such that the *resilience gain* (i.e., the number of followers) is maximized. As shown in Figure 2, we check the coverage of followers on Gowalla by greedily anchoring the vertices according to resilience gain, coreness gain [35], betweenness centrality [24] and closeness centrality [15], respectively. The result shows a clear gap in follower coverage among the resilience gain and other

Figure 1: Node engagement and coreness on Gowalla

Figure 2: Follower coverage of anchoring on Gowalla

metrics. It also indicates that the engagement enhancement is quite biased when applying the coreness gain, i.e., only a small portion of users benefit even if the budget is relatively large. As network resilience considers the engagement dynamic of all the users, the resilience gain adopted in the FM problem is more promising.

**Challenges and Contributions.** To the best of our knowledge, we are the first to study the FM problem to optimize network resilience via vertex anchoring. We prove the problem is NP-hard and W[2]-hard parameterized by the budget $b$. For problem approximability, it is NP-hard to approximate within an $O(n^{1-\epsilon})$ factor. Although GAC proposed in [35] can be migrated to solve the FM problem by modifying the computation of coreness gain to resilience gain, it is time-consuming in practice especially on large datasets due to large search space and unspecific design of techniques, e.g., it takes more than three days on the LiveJournal dataset when $b = 100$.

To efficiently solve the FM problem, we propose a series of novel computing techniques: (a) We first propose AdvGreedy, an advanced greedy approach with high efficiency. The key idea of AdvGreedy is to compute the followers of each candidate vertex based on shell component, which is more fine-grained than the core component tree used in GAC; (b) For the time-consuming follower computation, we propose a novel *explore-and-retract* strategy in local core decomposition, which can scan as few candidate followers as possible for accelerating the algorithm; (c) In addition, we propose a tight upper bound to reduce the number of vertices that require exact follower computation; and (d) We further refine the upper bound of follower number by combining it with the reuse technique.

Although our proposed AdvGreedy significantly outperforms GAC in both running time and resilience gain (shown in Section 7), it does not have any approximate guarantees (Corollary 2). To bridge this gap, we propose a time-dependent framework equipped with AdvGreedy, which applies branch and bound searching to obtain a high-quality solution quickly and then continues exploring promising spaces to produce better answers.

In summary, the main contributions of this paper are as follows.

- Motivated by existing studies, we propose and study the follower maximization (FM) problem to optimize network resilience. We prove the problem is NP-hard, W[2]-hard with respect to the budget parameter, and NP-hard to approximate within an $O(n^{1-\epsilon})$ factor.
- We design an advanced heuristic (AdvGreedy), which consists of three phases: upper bound computation, greedy selection, and reuse of intermediate results.
- To bridge the problem of approximate guarantee, we propose a novel time-dependent framework on the budget minimization problem of FM, which equips with AdvGreedy.

- Extensive experiments show that (i) our AdvGreedy is more effective than other heuristics including GAC in improving resilience gain; (ii) AdvGreedy is faster than GAC by more than 1 order of magnitude; and (iii) the time-dependent framework continues to produce better results over time.

## 2 RELATED WORK

The k-core [39, 45] has been extensively studied across various application scenarios, such as social networks [18, 27, 34, 48], web networks [11, 19], biological networks [2, 7], software networks [55], ecological networks [41], and financial networks [9].

To measure the ability of a network for withstanding and sustaining disturbances, extensive studies are conducted on network resilience/robustness. As surveyed in [20, 36, 42], there are various resilience measures based on different graph characteristics, e.g., adjacency, connectivity, distance, etc. The intuition of adjacency approaches is that nodes with many connections are more critical to the overall graph structure, including degree measures [22, 23], centrality [26, 32, 49]. Connectivity metrics measure the robustness of connecting/disconnecting the graph with key nodes [4, 12, 28]. Distance metrics consider the path length between node pairs, e.g., diameter-based metrics [1, 5]. Measures of network resilience/robustness vary on applications, e.g., Zitnik et al. [60] show that the connectivity-based metric can model the evolution of resilience in protein interaction networks. Nevertheless, the focuses of the above studies are different from our model, e.g., the centrality measures essentially consider the resilience on information flow [32, 49] while our FM problem is built on the true engagement dynamics of all the vertices.

As the coreness metric is the "best practice" for measuring the vertex engagement with network topology [38], many previous works consider measuring user engagement (network stability) by monitoring $k$-core structure, e.g., maximize the size of $k$-core via vertex anchoring [6, 13, 31, 37, 52], minimizing the size of $k$-core via vertex removal [53, 57], and edge manipulations [25, 40, 58, 59]. As the focus on the $k$-core with a fixed $k$ value is relatively a local view on user engagement, existing studies tend to consider the overall coreness dynamic of all the nodes, i.e., the overall coreness gain/loss [35, 51]. However, as shown in the introduction, the engagement enhancement is biased on certain nodes and the optimization of network resilience aims to enhance as many nodes as possible. Dey et al. [17] propose TMCV problem to maximize the number of coreness-changed vertices by deleting at most $b$ vertices. This is to consider the protection of tender nodes which is different from the enhancement of vertex anchoring in our FM problem.

## 3 PRELIMINARIES

We consider a simple, undirected and unweighted graph $G = (V, E)$, where $V(G)$ (resp. $E(G)$) represents the set of vertices (resp. edges) in $G$. We denote $n = |V(G)|$, $m = |E(G)|$ and assume $m > n$. Let $G[V]$ denote the induced graph by the vertex set $V$. Given a vertex $v$ in a subgraph $S$ of $G$, $N(v, S)$ denotes the neighbor set of $v$ in $S$, i.e., $N(v, S) = \{u \mid (u, v) \in E(S)\}$. The degree of $v$ in subgraph $S$, i.e., $|N(v, S)|$, is denoted by $d(v, S)$.

**Definition 1** ($k$-**core** $C_k(\cdot)$). Given a graph $G$ and an integer $k$, a subgraph $S$ is a $k$-core of $G$, if (i) each vertex $v \in S$ has at least

$k$ neighbors in $S$, i.e., $d(v, S) \geq k$; and (ii) $S$ is maximal, i.e., any supergraph of $S$ is not a $k$-core except $S$ itself.

Note that $k$-core in this paper is not required to be connected as in [39, 45], we use $k$-core to represent all the subgraphs satisfying Definition 1. According to the definition of $k$-core, every vertex in the graph has a unique coreness value.

**Definition 2 (coreness $c(\cdot)$).** Given a graph $G$, the coreness of a vertex $u \in V(G)$, denoted by $c(u, G)$, is the largest $k$ such that $C_k(G)$ contains $u$, i.e., $c(u, G) = \arg\max_k u \in C_k(G)$.

A graph can be decomposed into a hierarchy where the vertices are distinguished and arranged by their coreness values.

**Definition 3 (core decomposition).** Given a graph $G$, core decomposition is to compute the coreness of every vertex in $V(G)$.

The core decomposition can be computed in $O(m)$ time by recursively removing the vertex with the smallest degree in the remaining graph, and updating the degrees of its neighbors by bin sort [3].

In this paper, once a vertex $x$ in graph $G$ is **anchored**, its degree is considered as positive infinity while its neighbor set is not changed. Every anchored vertex is called an **anchor** or an **anchor vertex**. An anchor will never be removed in core decomposition of $G$, and core decomposition with anchors can still be computed in $O(m)$ time.

The existence of anchors may raise the corenesses of other vertices in the core decomposition with anchors. Let $c^A(u)$ denote the coreness of each vertex $u$ with the anchor set $A$. For each new anchor $x$, the coreness-increased vertices whose corenesses are not changed by previous anchors are the **followers** of $x$, i.e., the follower set of $x$ is $\{u \mid c^{A \cup \{x\}}(u) > c^A(u) \wedge c^A(u) = c(u)\}$, where $A$ is the anchor set before anchoring $x$. We assume the coreness of each anchor $x$ is increased by the anchoring, i.e., $c^{A \cup \{x\}}(x) > c^A(x)$.

**Definition 4 (resilience gain $g(\cdot)$).** Given a graph $G$ and the anchor set $A$, the resilience gain of $G$ regarding $A$, denoted by $g(A, G)$, is the number of followers by anchoring $A$, i.e., the number of vertices with coreness increased after anchoring $A$. We have $g(A, G) = \left| \{v \in V(G) \mid v \in A \vee c^A(v) > c(v)\} \right|$.

**Definition 5 (follower maximization problem).** Given a graph $G$ and a budget $b$, the follower maximization (FM) problem aims to find a set $A$ of at most $b$ vertices in $G$, such that the resilience gain regarding $A$, i.e., $g(A, G)$, is maximized.

The state-of-the-art solution for the FM problem is to adapt the GAC algorithm [35]. The main idea is to replace the coreness gain with the resilience gain in *greedy anchor selection* and *upper bound pruning*. More details are given in the appendix (Section A.1).

## 4 PROBLEM ANALYSIS

In this section, we first prove the FM problem is NP-hard and hard to approximate in general graphs, i.e., there is no polynomial time algorithm to approximate the optimal solution within a factor of $O(n^{1-\epsilon})$, for every positive constant $\epsilon > 0$.

**LEMMA 1.** *It is NP-hard to distinguish between instances of the FM problem where the optimal solution has value $\Omega(n)$ versus when the optimal solution has value $O(b)$.*

PROOF SKETCH. We prove the lemma through a reduction from the *set cover decision* (SCD) problem [29] to our FM problem, by constructing corresponding FM instances for any general SCD instances. The main idea of designing new FM instances is to construct graphs, which can get $O(b)$ resilience gain when the corresponding SCD instance is a *no-instance*, and can get $\Omega(n)$ resilience gain when the corresponding SCD instance is a *yes-instance*. Since the SCD problem is NP-complete, Lemma 1 can be proved. □

**COROLLARY 1.** *Given a graph $G$, the FM problem is NP-hard.*

PROOF. In the rest of the paper, please find the proofs in Section A.5 for all the theorems/corollaries. □

Lemma 1 further immediately indicates that there does not exist any $O(n^{1-\epsilon})$ approximate solution for the FM problem.

**COROLLARY 2.** *For any $\epsilon > 0$, it is NP-hard to approximate the FM problem on general graphs within an $O(n^{1-\epsilon})$ factor.*

From a parameterized perspective, we prove that the FM problem is W[2]-hard with respect to the budget parameter $b$.

**THEOREM 3.** *The FM problem is W[2]-hard parameterized by $b$.*

Besides, we prove the properties of the resilience gain function.

**THEOREM 4.** *Resilience gain $g(\cdot)$ is monotonic but not submodular.*

## 5 AN ADVANCED GREEDY APPROACH

As mentioned in Section A.1, GAC [35] finds the followers of each anchor vertex $x$ in those $k$-core components which contain at least one neighbor of $x$ with the same or higher coreness as $x$. Better still, it develops reuse techniques and upper bound based on the $k$-core component. However, $k$-core component is not the atomic unit in finding the followers. We can find that if a vertex $u$ is a follower of an anchor $x$, then there exists at least one path from $x$ to $u$ (Lemma 2) s.t. all the vertices in the path except $x$ share the same coreness. Motivated by this, we propose the concept of shell components, which are connected subgraphs of $k$-core components with the same $k$ and contain all the followers. Therefore, we can efficiently find all the followers of an anchor in the smaller shell components rather than in the larger $k$-core components. Based on the shell component structure, we propose our approach AdvGreedy. Intuitively, AdvGreedy outperforms GAC by following reasons: **(1) Follower computation.** Since each shell component is a subgraph of a $k$-core component, the search space is reduced significantly. Besides, we propose a *explore-and-retract* strategy to further reduce the number of scanned vertices, which utilizes a multi-queue data structure to maintain the scan order. **(2) Reuse results.** The shell component is more fine-grained than the $k$-core component. Consider a vertex $u$ whose shell component remains unchanged while $k$-core component has changed, thus $u$'s follower results can be reused in AdvGreedy while needs re-computation in GAC. **(3) Upper bound computation.** We utilize a shell component to tighten the upper bound of the follower numbers, significantly improving the pruning effect of non-candidates.

In what follows, we first introduce the *shell component* structure (Section 5.1), and combine it with the explore-and-retract strategy to propose the followers computation method (Section 5.2). We

---

**Algorithm 1: ShellDecomp($G$)**

**Input** : $G$ : the graph
**Output** : $\mathcal{SC}$ : the index of shell components in $G$
1 Compute $c(u, G)$ of each $u \in V(G)$ by core decomposition [3];
2 **for** each *unassigned* $u \in V(G)$ **do**
3     $G' \leftarrow$ the connected subgraph of $H_{c(u,G)}(G)$ which contains $u$;
4     $S \leftarrow$ a new shell component;
5     $S.c \leftarrow c(u,G); S.V \leftarrow V(G'); S.E \leftarrow E(G')$;
6     $v$ is set *assigned* **for each** $v \in V(G')$;
7     $\mathcal{SC}[v] \leftarrow S$ **for each** $v \in V(G')$;

8 **return** $\mathcal{SC}$

---

then detail the mechanism to reuse the intermediate results across greedy interactions (Section 5.3) and the design of the upper bound pruning method (Section 5.4). Finally, we put the above techniques together and propose our AdvGreedy algorithm (Section 5.5).

## 5.1 Shell Component Structure

**Definition 6 ($k$-shell).** Given a graph $G$ and a positive integer $k$, the $k$-shell, denoted by $H_k(G)$, is the set of vertices in $G$ with their corenesses exactly equal to $k$, i.e., $H_k(G) = V(C_k(G)) \setminus V(C_{k+1}(G))$.

**Definition 7 (shell component).** Given a graph $G$ and the $k$-shell $H_k(G)$, a subgraph $S$ is a shell component of $H_k(G)$, if $S$ is a maximal connected component of the induced subgraph $G[H_k(G)]$.

Different from Definition 9, where vertices in the same $k$-core component can be connected through other vertices whose coreness is larger than $k$, vertices in the same shell component must be connected through vertices that share the same coreness $k$. A $k$-shell is formed by the vertices in a series of non-overlapping shell components, and each vertex is contained in exactly one shell component. Note that in core decomposition, the deletion sequence of the shell components of $H_k(G)$ can be arbitrary. For a shell component $S$ of $H_k(G)$, we denote $S.V$, $S.E$ and $S.c$ as the vertex set, edge set and coreness of any vertex in $S$ respectively, i.e., $S.V = V(S)$, $S.E = E(S)$ and $S.c = k$. We use structure $\mathcal{SC}$ to index the shell components for all the vertices. For each $v \in V(G)$, $\mathcal{SC}[v]$ is the only shell component that $v \in \mathcal{SC}[v].V$.

Vertices in each shell component can be further divided into different vertex sets, named **layers**, according to their deletion sequence in core decomposition [3]. We use $l(u)$ to denote the layer number of vertex $u$, and use $H^i$ to denote the $i$-layer vertex set in the $k$-shell $H_k(G)$, i.e., the set of vertices that are deleted in the $i$-th batch of $H_k(G)$ in core decomposition. Formally, $H^i = \{u | d(u, G_i) < k + 1 \wedge u \in H_k(G)\}$, where $G_1 = C_k(G)$, and for $i \geq 1$, $G_{i+1}$ is the induced subgraph of vertex set $V(G_i) \setminus H^i$, i.e., the deletion of the $i$-th layer will produce the $(i+1)$-th layer. The layer of each vertex can be computed easily during the core decomposition. For each vertex $u$, we denote the **coreness-layer pair** of $u$ as $\mathcal{P}(u)$, i.e., $\mathcal{P}(u) = (c(u), l(u))$. We then define the order of the coreness-layer pair, $\mathcal{P}(u) \prec \mathcal{P}(v)$ iff $c(u) < c(v)$ or $c(u) = c(v) \wedge l(u) < l(v)$.

**Shell Component Computation.** Algorithms 1 illustrates the decomposition of each vertex into its shell component, which costs $O(m)$ time. We first conduct core decomposition on $G$ and get the coreness of each vertex (Line 1). Then we traverse all *unassigned*

vertices to construct all the shell components (Lines 2-8). Each time visit an *unassigned* vertex $u$, we first apply BFS to get the connected subgraph $G'$ of $(c(u, G))$-shell which contains $u$ (Line 3), then create a new shell component $S$ (Line 4-5) and mark vertices in $V(G')$ as *assigned* (Line 6). After that, we set $\mathcal{SC}[v]$ for $v \in V(G')$ by $S$ (Line 7). When all the vertices are set *assigned*, we can get $\mathcal{SC}$ (Line 8).

## 5.2 Follower Computation on Shell Component

From Lemma 3, we can compute the resilience gain by computing the number of followers when adding a new anchor. By the definition of $k$-core, we know that if the coreness of a vertex $v$ increases to $c(v) + 1$, $v$ must have at least $c(v) + 1$ neighbors whose corenesses are at least $c(v) + 1$, and we call these neighbors **supporters** of $v$.

For follower computation, [35] further define the upstair path and limit the candidate followers (search space) based on it.

**Definition 8 (Upstair Path).** An upstair path in $G$ for $u \in V(G)$ w.r.t a given anchor $x$ if there is a path $x \rightsquigarrow u$ where (i) for every vertex $y(y \neq x)$, $c(y) = c(u)$; and (ii) for every two consecutive vertices $v$ and $v'$ from $x$ to $u$, $(v, v') \in E(G)$ and $\mathcal{P}(v) \prec \mathcal{P}(v')$.

**Lemma 2 ([35]).** A vertex $u \in V(G)$ is a follower of the anchor $x$ implies that there is an upstair path $x \rightsquigarrow u$ in $G$.

Benefiting from shell components, we extend Lemma 2 to following theorem to limit candidate followers of an anchor. Let $SN(v)$ denote **successive neighbors** of $v$ (neighbors with higher coreness-layer pairs), i.e., $SN(v) = N(v, G) \cap \{w \mid \mathcal{P}(v) \prec \mathcal{P}(w)\}$.

**Theorem 5.** A vertex $v \in V(G)$ is a follower of vertex $x$ implies that $v \in \bigcup_{S \in CS(x)} S.V$, where $CS(x) = \bigcup_{u \in SN(x)} \mathcal{SC}[u]$.

According to Theorem 5, we use shell components as the basic units to compute the followers of each anchor $x$. We then show that the follower computation can be conducted on each shell component independently. The increase of $v$'s coreness after anchoring $x$ must be caused by the increased number of $v$'s supporters. Denoted by $u$, $v$'s supporters can be divided into three sets: (1) $u \in A$, the anchors can always support its neighbors; (2) $c^A(u) > c^A(v)$, since the coreness of $v$ increases at most 1, $u$ is still a supporter of $v$ after anchoring $x$; (3) $c^A(u) = c^A(v)$, $u$ will remain as a supporter of $v$ if $c(u)$ also increases after anchoring $x$.

Since case (1) is easy to identify, we focus on the latter two. For a vertex $v$, if its coreness increases after anchoring $x$, it is likely that new supporters of $v$ come from its neighbors with the same coreness before anchoring $x$. In this case, to determine whether a vertex $v$ is a follower of $x$, we only need to focus on $v$'s neighbors who are in $\mathcal{SC}[v]$ before anchoring $x$. As a result, we can compute $x$'s followers on each of its candidate shell components $S \in CS(x)$ independently.

**Explore-and-Retract Strategy.** To compute followers of anchor $x$, we employ the explore-and-retract strategy to check if the corenesses of vertices who lie on any upstair path from $x$ will increase. Specifically, we continue to *explore* the higher-layer neighbors (due to Lemma 2) of the vertex which we suppose its coreness will probably increase, and immediately *retract* when meeting a vertex whose coreness cannot increase, to check whether this "impossible" vertex will cause its lower-layer neighbors to also become "impossible".

As shown in Algorithm 2, we first find candidate shell components $CS(x)$ based on Theorem 5 (Lines 1-2). For each $S \in CS(x)$,

---

**Algorithm 2: FindFollowers($x, G, \mathcal{SC}$)**

**Input** : $x$ : the anchor, $G$ : the graph, $\mathcal{SC}$ : the shell components

**Output** : $F[x][\cdot]$: shell component classified follower sets of $x$

1   $SN(x) \leftarrow N(x, G) \cap \{w \mid \mathcal{P}(x) \prec \mathcal{P}(w)\}$;

2   $CS(x) \leftarrow \bigcup_{u \in SN(x)} \mathcal{SC}[u]$;

3   **for** each $S \in CS(x)$ **do**

4      **if** $S \in ReuseSC(x)$ **then continue**;    // Section 5.3

5      $F[x][S] \leftarrow \varnothing$;

6      $max\_layer \leftarrow \max_{v \in S.V} l(v)$;

7      Initialize queues $Q_1, \cdots, Q_{max\_layer}$;

8      Push $v$ into $Q_{l(v)}$ for each $v \in S.V \cap SN(x)$;

9      **for** $i \leftarrow 1$ to $max\_layer$ **do**

10        **while** $Q_i$ is not empty **do**

11          $v \leftarrow Q_i.front()$; $Q_i.pop()$;

12          $SN(v) \leftarrow N(v, G) \cap \{w \mid \mathcal{P}(v) \prec \mathcal{P}(w)\}$;

13          $sup(v) \leftarrow |N(v, G) \cap (Q_i \cup SN(v) \cup F[x][S] \cup \{x\})|$;

14          **if** $sup(v) \geq c(v) + 1$ **then**

15            $F[x][S] \leftarrow F[x][S] \cup \{v\}$;

16            Push $u$ into $Q_{l(u)}$ for each $u \in SN(v) \cap S.V$;

17          **else**

18            Initialize queue $Q$ and push $v$ into $Q$;

19            **while** $Q$ is not empty **do**

20              $u \leftarrow Q.front()$; $Q.pop()$;

21              $F[x][S] \leftarrow F[x][S] \setminus \{u\}$;

22              **for** each $w \in N(u, G) \cap F[x][S]$ **do**

23                $sup(w) \leftarrow sup(w) - 1$;

24                **if** $sup(w) \leq c(w)$ **then** push $w$ into $Q$;

25   **return** $F[x][\cdot]$;

---

**Algorithm 3: Reuse($x, G, A, \mathcal{SC}$)**

**Input** : $x$ : the anchor, $G$ : the graph, $A$ : the anchor set, $\mathcal{SC}$ : the shell components of $G$

**Output** : $ReuseSC(v)$ for each non-anchor vertex $v$, where $F[v][S]$ can be reused for each $S \in ReuseSC(v)$

1   **for** each $v \in V(G) \setminus A$ **do**

2      $ReuseSC(v) \leftarrow CS(v)$;

3      Remove $S$ from $ReuseSC(v)$ if $F[v][S]$ is not computed;

4   $V^* \leftarrow \bigcup_{S \in CS(x)} S.V$;

5   Compute $c'(\cdot)$ through core decomposition [3];

6   $\mathcal{SC}' \leftarrow \textbf{ShellDecomp}(G)$;

7   $S'^* \leftarrow \bigcup_{v \in V^*} \mathcal{SC}'[v]$; $V'^* \leftarrow \bigcup_{S \in S'^*} S.V$;

8   Remove $S \in \mathcal{SC}$ from all $ReuseSC(\cdot)$ if $V'^* \cap S.V \neq \varnothing$;

9   **return** $ReuseSC(v)$ for each $v \in V(G) \setminus A$;

---

if $x$'s followers in $S$ remain the same as the last iteration, we reuse the results (Line 4, detailed in Section 5.3). Otherwise, we find its followers in each component $S$ independently (Lines 5-25), which are maintained in $F[x][S]$ (Line 5). To apply the *explore-and-retract strategy*, we scan vertices in ascending order of their coreness-layer pairs, and decide whether the coreness of a vertex will increase by checking its supporter number. To organize candidate followers in linear time, we construct multiple queues for different layers. More specifically, for each $S$, we use a sequence of queues $\{Q_1, Q_2, \cdots, Q_{max\_layer}\}$ to maintain the traverse order, where $max\_layer$ denotes the maximum layer number in $S$ (Lines 6-7). We first push $x$'s successive neighbors in $S$ into the queues (Line 8), then traverse each element $v$ in $Q_i$ in ascending order of $i$ (Lines 9-11).

For each vertex $v$, we denote its supporter number as $sup(v)$ and divide its neighbors $u \in \mathcal{SC}[v]$ into three categories to compute $sup(v)$ : (i) *unexplored* and $l(u) \geq l(v)$: We first assume that $u$ is a supporter of $v$. If $sup(v) \geq c(v) + 1$, we regard $v$ as a potential coreness-increased vertex and will explore $u$ later. If we later find that $u$'s coreness cannot increase (i.e., $u$ is actually not a supporter of $v$), we perform *retract* to check if $v$'s coreness will increase. (ii) *unexplored* and $l(u) < l(v)$: In this case, $u$ cannot be a supporter of $v$. As we scan vertices in layer order, $u$ will never be explored. (iii) *explored*: We have temporarily decided whether the coreness of $u$ will increase. If so, $u$ can be a supporter of $v$. Thus $sup(v)$ includes (i) $Q_i \cup SN(v)$, (iii) $F[x][S]$ and $x$ (Lines 12-13). We then check if

$v$'s coreness can increase (Line 14). If so, we temporarily assume $v$ is the follower of $x$, put it into $F[x][S]$ (Line 15) and continue to *explore* its higher-layer neighbors (Line 16). Otherwise, we ensure that $v$'s coreness will not increase, then recursively *retract* to check if other vertices will remain in $F[x][S]$. Vertex that does not satisfy the coreness-increasing requirement will be removed from $F[x][S]$ (Line 21). As the removed vertex may have been regarded as a supporter of its neighbors before, we recursively *retract* to check its neighbors' supporter numbers (Lines 22-24). Therefore, the final remaining vertices in $F[x][\cdot]$ are the true followers of $x$ (Line 25).

The time complexity of Algorithm 2 is $O(m)$, because each edge is accessed at most two times: *explore/retract* when meeting/failing the coreness-increasing requirement. In practice, the number of scanned vertices in Algorithm 2 is much smaller, as the explore-and-retract strategy will make local decomposition early terminate.

**Example 1.** We explain an example of applying Algorithm 2 to compute the follower set of $x = v_1$ in the graph of Figure 3. We first push $x$'s neighbors $v_2$ and $v_6$ in turn into $Q_2$. For $v_2$, we have $sup(v_2) = 3 \geq c(v_2) + 1$, because $v_1 \in \{x\}$, $v_5 \in SN(v_2)$ and $v_6 \in Q_2$. Thus we push $v_2$ into the follower set $F[x][S]$ and push $v_5$ into $Q_3$. For $v_6$, since the layers of $v_7$ are less than that of $v_6$, we have $sup(v_6) = 2 < c(v_6) + 1$, which triggers the retract strategy. It makes $sup(v_2)$ decrease and turns back to check the supporter number of $v_2$. We find that $sup(v_2) = 2 < c(v_2) + 1$, which means $v_2$ is actually not a follower of $x$ and will be removed from $F[x][S]$. Then we enumerate the elements in $Q_3$. For $v_5$, we have $sup(v_5) = 2 < c(v_5) + 1$, and there is no more element in the queues, thus the Algorithm 2 terminates and the follower set of $v_1$ is empty.

## 5.3 Reuse Follower Computation Results

The greedy algorithm contains $b$ iterations, and we apply the reuse technique in order to avoid redundant computations. For each vertex $v \in V(G) \setminus (A \cup \{x\})$ and each shell component $S$, we decide whether the computed follower $F[v][S]$ will remain the same after anchoring $x$, thus can be reused in the next selection iteration.

Algorithm 3 finds all the candidate anchors and shell components in which the follower number can be reused. For each $v \in V(G) \setminus A$, $ReuseSC(v)$, initialized as $CS(v)$, contains all $v$'s candidate followers (Lines 1-2, Theorem 5). For each $S \in ReuseSC(v)$, $F[v][S]$ must have been computed before (Line 3). Let $V^*$ denote the vertex set of all

Figure 3: Shell component example for techniques

Figure 4: Solution tree of the graph in Figure 3

shell components in $CS(x)$ (Line 4), we update the coreness after anchoring $x$ and construct new shell components (Lines 5-6). Let $V'^*$ denote the vertex set of all new shell components containing some vertex in $V^*$ (Line 7). Original shell components which contain some vertex in $V'^*$ can not be reused, hence are removed from $ReuseSC(\cdot)$ (Line 8). Algorithm 3 runs in $O(m)$ time complexity as we will scan each edge at most once to get $ReuseSC(\cdot)$ initially, and core decomposition and Algorithm 1 both needs $O(m)$ time.

THEOREM 6. *(Correctness). After anchoring $x$, for every non-anchor vertex $v$, we have $F[v][S]$ remaining the same if $S \in ReuseSC(v)$.*

### 5.4 A Tighter Upper Bound

We first review the upper bound pruning used in GAC. Based on Lemma 2, Linghu et al. propose the upper bound of follower number of any non-anchor vertex $x$, i.e., $UB_\sigma(x) = 1 + \sum_{u \in SN(x)} UB_i(u)$, where $UB_i(x) = 1 + \sum_{u \in SN(x) \cap \{v|c(v)=c(x)\}} UB_i(u)$. However, the following example shows this bound has much overlap.

**Example 2.** Consider computing the upper bound of $v_5$ in the graph of Figure 3. We have $UB_i(v_4) = UB_i(v_9) = 1$ and then $UB_i(v_3) = UB_i(v_8) = 1 + UB_i(v_4) + UB_i(v_9) = 3$. Thus $UB_\sigma(v_5) = 1 + UB_i(v_3) + UB_i(v_8) = 7$, which double counts $v_4$ and $v_9$.

Worse still, we experimentally find that a large ratio of the upper bounds computed in this way exceeds $n$ (shown in Table 3). To refine the technique, according to Lemma 2, we first consider the size of the **upstair DAG** as the direct upper bound, i.e., the number of vertices that can be reached from $x$ through any upstair path. However, there exists no linear algorithm which can compute the exact size of the reachable DAG for each vertex [14]. We thus refine the upper bound based on the shell components. Specifically, for an candidate anchor $x$, we first get the upper bound of its followers of each shell component $S$, making it no more than the number of vertices with larger layers than $x$ in $S$, i.e., $UB(x, S) = \min \{|S.V \cap U(x)|, \sum_{u \in SN(x) \cap S.V} UB(u, S)\}$, where $U(x) = \{v \mid \mathcal{P}(x) \prec \mathcal{P}(v)\}$. If $x$'s coreness has never changed before, we set $UB(x, \mathcal{SC}[x]) = UB(x, \mathcal{SC}[x]) + 1$. Then the upper bound of $x$'s followers is $UB(x) = \sum_{S \in CS(x)} UB(x, S)$. Furthermore, applying the reuse technique, if the follower result $F[x][S]$ can be reused in current iteration, we can use it directly as it is exactly the number of $x$'s followers, i.e., the tightest upper bound.

THEOREM 7. *Given a graph $G$, a current anchor set $A$ and a vertex $x \in V(G) \setminus A$, we have $g(A \cup \{x\}, G) - g(A) \le UB(x)$.*

As we can compute the upper bounds of all the candidate anchors in a reverse order of topological sorting of their coreness-layer pairs, the time complexity of the upper bound computation is $O(m)$.

---

**Algorithm 4: AdvGreedy($G, b$)**

**Input** : $G$ : the graph, $b$ : number of anchors
**Output** : $A$ : the set of anchored vertices

1 Compute $c[\cdot]$ through core decompostion [3];
2 $\mathcal{SC} \leftarrow$ **ShellDecomp**($G$);
3 $g \leftarrow \varnothing$;
4 **for** $i \leftarrow 1$ to $b$ **do**
5    $x \leftarrow null$; $\Delta \leftarrow 0$;
6    Compute *upper bounds* $UB[u]$ **for** each $u \in V(G) \setminus A$;
7    **for** each $u \in V(G) \setminus A$ with decreasing order $UB(u)$ **do**
8      **if** $UB(u) > \Delta$ **then**
9        $F \leftarrow$ **FindFollowers**($u, G, \mathcal{SC}$);
10        **if** $|F \setminus g| > \Delta$ **then**
11          $\Delta \leftarrow |F \setminus g|$; $x \leftarrow u$;
12      **else Break**;
13    $A \leftarrow A \cup \{x\}$; $d(x) \leftarrow +\infty$;
14    **Reuse**($x, G, A, \mathcal{SC}$);
15 **return** $A$;

---

### 5.5 An Advanced Greedy Approach

Algorithm 4 shows the details of our final AdvGreedy algorithm which combines all the techniques proposed in the last 4 Sections. We first compute the coreness of each vertex and construct the shell components in $G$ (Lines 1-2). Let $g$ be the set of vertices whose coreness has changed (Line 3). In each iteration of the greedy heuristic, $x$ records the best anchor found so far, and $\Delta$ records its resilience gain (Line 5). We first compute the follower upper bound of each candidate anchor $u$ in a reverse order of the topological sorting of their coreness-layer pairs (Line 6). Then, we enumerate each candidate anchor in a decreasing order of their follower upper bounds (Line 7), and compute its exact follower set to update $x$ and $\Delta$ when necessary (Lines 8-12). Note that in the follower computation, we need to remove vertices whose corenesses have already increased before from the follower set, since they can not make additional contributions to the resilience gain. When we determine the best anchor $x$ in the current iteration, we update anchor set $A$ and set degree of $x$ as infinity (Line 13). We then compute shell components which can be reused in the next iteration for each vertex (Line 14). After $b$ iterations, Algorithm 4 returns the anchor set $A$ (Line 15).

**Budget Minimization Problem.** Algorithm 4 can be readily adapted to solve the budget minimization problem of FM. Specifically, the input budget of AdvGreedy is replaced with the target resilience gain $g'$, and the termination condition is set as when the current resilience gain $g$ with the anchor set $A$ is no less than $g'$, i.e., $g(A) \ge g'$.

## 6 A TIME-DEPENDENT FRAMEWORK

Since the FM problem is NP-hard to approximate within an $O(n^{1-\varepsilon})$ factor, it is hard to develop an efficient algorithm with a theoretical approximate guarantee. To bridge this gap between theory and practice, we propose an algorithmic paradigm in this section, which can be instantiated to output a good solution quickly and then look for better solutions within the given time limit based on AdvGreedy.

Specifically, we design an exact algorithm paradigm for the budget minimization problem and then consider returning to solve the

FM problem. The exact algorithm paradigm needs to explore all the possible $2^b$ solutions, which are encoded by a solution tree $\mathcal{T}$, i.e., $\mathcal{T}$ is a perfect binary tree with $2^b$ leaves. Every node in $\mathcal{T}$ has two children. Its left child means adding a new vertex $x$ into the anchor set $A$, while its right child means $x$ will not be considered as an anchor. We use $\mathcal{T}(A, A_\neg)$ to denote each tree node, where $A$ is the anchor set of the current node, and $A_\neg$ is the set of disregarded vertices up to now. For each tree node, the "to be decided vertex" $x$ is chosen by the greedy approach. Specifically, for a tree node $\mathcal{T}(A, A_\neg)$, the next vertex we choose to add into $A$ or $A_\neg$ is $x = \arg\max_{u \in V(G) \setminus (A \cup A_\neg)} g(A \cup \{x\}) - g(A)$, i.e., the left child node is $\mathcal{T}(A \cup \{x\}, A_\neg)$ and the right child node is $\mathcal{T}(A, A_\neg \cup \{x\})$. We apply a DFS to search for solutions in $\mathcal{T}$, thus the first solution we can find is the result from the greedy method, which satisfies *output a good solution quickly*. Then we will explore more vertices according to their follower numbers, which means the vertices that can lead to larger resilience gain will be explored first, this follows *search for better solutions as quickly as possible*.

**Reuse Intermediate Results.** As DFS has two main actions, continuing to *search* the child nodes and *backtrack* to the father nodes, we design a linear space implementation for reusing the intermediate results. Specifically, in the subtree rooted at $\mathcal{T}(A, A_\neg)$, we greedily add vertices into $A$ in the child nodes, store the follower upper bound and the reusable shell components for each vertex $v$ in $UB[|A|][v]$ and $ReuseSC[|A|][v]$. Then we push the follower results of vertices into $H[|A|]$, where $H[\cdot]$ is a max heap and is ordered by the follower number of each vertex. Thus we only need to compute $UB[|A|][\cdot]$ and $ReuseSC[|A|][\cdot]$ once in the subtree, and continue computing the followers of each vertex based on $H[|A|]$.

**Return to FM Problem.** The above search process deals with the budget minimization problem, we then introduce how to use the search results to further improve the solution of the FM problem. As the first result returned is the same as the greedy approach, we use the given budget to get the first result and use it as the target resilience gain. Then we continue to apply the paradigm to search for smaller budgets, and once we get a smaller budget, we naturally have more extra budgets to improve the resilience gain, i.e., we apply the greedy method to select $(b - b_{min})$ more anchors.

The detailed description and pseudo-code of the paradigm can be found in the appendix.

**Pruning Strategies.** To speed up the search of $\mathcal{T}$, we apply some effective pruning strategies. Recall that we turn to the budget minimization problem to further solve the FM problem. Let $b_{\min}$ denote the current best solution, and $g_t$ denote the target resilience gain, we apply the following strategies to accelerate the search process:

(1) If $g(A) \geq g_t$, the subtree can be pruned, for other solutions in it can not have smaller budgets.

(2) If $g(A) < g_t$ and $|A| \geq b_{\min} - 1$, the subtree can be pruned, because the best possible solution in its subtree is $b_{\min}$.

(3) If $g(V(G) \setminus A_\neg) < g_t$, the subtree can be pruned, since no solutions in the subtree can reach the target gain due to the monotonicity of $g(\cdot)$ (Theorem 4).

**Bounded-death Heuristic.** As the solution space is still large even with the above pruning techniques, to limit the search to a relatively better region in the solution space, we apply the bounded-death heuristic [56] in our framework. Specifically, we further prune the

subtree rooted at tree node $\mathcal{T}(A, A_\neg)$ if $|A_\neg| > \lambda$, where $\lambda \geq 0$ is a given constant integer. In our paradigm, if $\lambda = 0$, the result is exactly what the greedy approach AdvGreedy finds.

**Example 3.** Consider the graph in Figure 3, we construct its corresponding search tree in Figure 4 with budget $b = 2$ and $\lambda = 1$. In each tree node, we mark the current $A$ and $A_\neg$, and for each edge, we use $+v_i$ and $-v_i$ to denote adding $v_i$ into $A$ or $A_\neg$. For the root node, $A$ and $A_\neg$ are both originally $\varnothing$ and the greedy approach selects $v_7$ as the first anchor. For the left child of the root, it adds $v_7$ into $A$, and greedily selects the next anchor as $v_1$. For its left child, we further add $v_1$ into $A$ and get the first solution $A = \{v_1, v_7\}$. Then we find that we can prune the subtree rooted at the right child of node $(\{v_7\}, \varnothing)$, because $g(A) = 5 < g_t$ and $|A| = 1 \geq b_{\min} - 1$ (Pruning 2). For the right child of the root, it adds $v_7$ into $A_\neg$, thus due to $\lambda = 1$, the subtree rooted in its right child will be pruned. For the node $(\{v_2\}, \{v_7\})$, its left child's subtree is pruned due to Pruning 2, and its right child's subtree is pruned because of $\lambda = 1$.

## 7 EXPERIMENTAL EVALUATION

**Datasets.** The experiments are conducted on 8 public datasets. Wiki is from KONECT [30]. The other datasets are available from SNAP [33]. The statistics of datasets are included in the appendix, where the largest dataset in our experiments contains $3,072,441$ vertices and $117,185,083$ edges.

**Environments.** Experiments are performed on a CentOS Linux server (Release 7.5.1804) with Quad-Core Intel Xeon CPU (E5-2640 v4 @ 2.20GHz) and 128G memory. All algorithms are implemented in C++17. Source code is compiled by GCC under -O3 optimization.

### 7.1 Compared Methods

Towards effectiveness, we compare greedy method (AdvGreedy / GAC-FM) with exact ones and other 7 heuristics. We survey heuristics proposed in related works and adapt them to solve our problem.

**Vertex Attribute.** The basic heuristics are the attributes of vertices. Degree (Deg). Deg anchors $b$ vertices with the highest degree. Coreness (Core). Core anchors $b$ vertices with the highest coreness.

**Bound of Resilience Gain.** We can use the estimated bounds of resilience gain as another type of heuristics to select anchors. Upper Bound (UB). UB chooses $b$ vertices with the largest upper bound, i.e., $UB(x)$ for each vertex $x$ (details in Section 5.4). Upstair DAG Size (UD). UD chooses $b$ vertices with the largest upstair DAG size, i.e., the number of vertices that can be reached from each vertex through its upstair paths. It is the tighter version of UB, but it is time-consuming since there is no linear algorithm. Successive Degree (SD). Experiments in [35] compare with GAC by the successive degree, that is, choose $b$ anchors with the highest successive degree, i.e., $|SN(\cdot)|$. It can be regarded as a lower bound of the upper bound (1-hop of the upstair DAG).

**Score Function.** Applying scoring functions to evaluate the candidate vertices' quality is a common method in related works. The details of the following two algorithms are included in the appendix. Shapley Value (SV). Shapley Value is a concept in cooperative game theory. Motivated by [40], we design a Shapley Value to capture the importance of a vertex inside a vertex set.

Figure 5: Resilience gain from different heuristics when $b = 100$

(a) Overall running time

(b) Gowalla

(c) Youtube

Figure 6: Running time

(a) Gowalla

(b) Youtube

Figure 7: Performance of GreedySearch when budget=100

Combinational Score (CS). Motivated by the score function from [37], we consider the combinational effect of anchors and design a new heuristic for our problem.

## 7.2 Experimental Results

**Exp 1: Comparison with Other Heuristics.** We compare the resilience gain of AdvGreedy with other 7 heuristics (details in Section 7.1) when the budget is 100. Note that UD and SV do not return results within three days in three larger datasets, we mark them by "OOT" in the figure. As shown in Figure 5, AdvGreedy always performs the best among all the heuristics. CS and SV perform relatively well as they both consider the income of anchor combination. They may fail on some datasets, e.g., in Stanford the performance gap between CS and AdvGreedy is huge. The efficiency of SV is much worse than AdvGreedy even when reducing the samples. Among three bound heuristics, UD performs the best as it equips with a tighter bound. For vertex attributes, the performance of Deg is better than Core, but it is still much worse than AdvGreedy. Core performs the worst, since vertices with larger corenesses are originally others' supporters thus anchoring them will not provide additional support.

**Exp 2: Overall Efficiency.** Figure 6a shows the total running time of GAC-FM and AdvGreedy on all datasets when $b = 100$. GAC-FM cannot return results on Orkut within one week, thus we mark it as "OOT". In all 8 datasets, AdvGreedy always outperforms GAC-FM by almost 1 order of magnitude and up to 2 orders. Besides, the gap becomes larger with the scale of the datasets increasing.

**Exp 3: Varying the Budget.** Figures 6b and 6c present the running time on Gowalla and Youtube when budgets vary from 1 to 1000.

As GAC-FM do not return results within 24 hours when $b \geq 487$, we do not report its running time in Figure 6c. In both two figures, the slope of the curve decreases as the budget increases, indicating that AdvGreedy has excellent scalability when the budget is large. Besides, AdvGreedy is always faster than GAC-FM by more than 1 and 2 orders of magnitude in Gowalla and Youtube, respectively. We can also find that the gap between them is huge when the budget is relatively small because of the refined upper bound.

**Exp 4: Performance of Time-Dependent Framework.** Figures 7a and 7b show the performance of the time-dependent search framework on Gowalla and Youtube when $b = 100$ respectively. The framework first finds a resilience gain of 4026 on Gowalla in 188s and 4578 on Youtube in 846s, similar to AdvGreedy. GreedySearch continues to search for a better solution with parameter $\lambda$ varying from 1 to 3. The framework can always discover better solutions as the search time increases. The performance is the best when $\lambda = 2$ on both datasets, as smaller $\lambda$ may result in excessive pruning, and bigger $\lambda$ may be time-consuming, e.g., GreedySearch can not terminate within $10^6$ seconds, when we set $\lambda = 3$ on Gowalla.

## 8 CONCLUSION AND FUTURE WORK

In this paper, we propose and study the follower maximization problem, aiming to maximize coreness-increased vertices by finding an anchor set. We prove the problem is NP-hard, and NP-hard to approximate within a factor of $O(n^{1-\epsilon})$. The problem is also W[2]-hard parameterized by budget. Given such hardness, we develop an efficient greedy method AdvGreedy based on shell components and pruning techniques. Extensive experiments on 8 real-life networks demonstrate the effectiveness of AdvGreedy, especially on massive graphs. To bridge the gap between theory and practice, a time-dependent framework is proposed, producing a solution quickly and continuing to search for better solutions if time permits. In future work, it is promising to design more powerful heuristics which can achieve similar effectiveness while more efficient, then the extended generic framework may beat the greedy approach on both sides.

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

# A APPENDICES

## A.1 State-of-the-Art from Existing Method

Linghu et al. [35] propose a greedy algorithm GAC for the anchored coreness problem. In a nutshell, GAC starts from an empty anchor set $A = \varnothing$, and then iteratively finds one best anchor $u$ with the highest coreness gain to add into $A$ in each of the $b$ iterations, i.e.,

$$u = \arg\max_{v \in V(G) \setminus A} \left( cg(A \cup \{v\}, G) - cg(A, G) \right),$$

where $cg(A, G)$ is the coreness gain of anchor set $A$ in $G$. The computation of $cg(\cdot)$ in GAC is mainly based on the following lemma.

LEMMA 3 ([35]). *If a vertex $x$ is anchored in $G$, the coreness of any $u \in V(G) \setminus \{x\}$ will either not decrease or increase by at most 1.*

We can apply a core decomposition to compute the coreness gain for each candidate vertex, which requires $O(m)$ time. The greedy method is conceptually simple, while it is computationally expensive. GAC speeds up its efficiency by utilizing the *core component tree*, widely used in related works [35, 39, 43, 44, 51], which organizes $V(G)$ based on the $k$-core components in $G$ with different $k$.

**Definition 9** ($k$-**core component**). Given a graph $G$ and the $k$-core $C_k(G)$, a subgraph $S$ is a $k$-core component if $S$ is a connected component of $C_k(G)$.

Linghu et al. find that the followers of each vertex $x$ are constrained to the core components that contain at least one neighbor of $x$ with the same or higher coreness as $x$, and these components are denoted by $\mathcal{TC}(x)$. Therefore, GAC finds followers of each vertex through partial core decomposition, i.e., applies core decomposition independently in components in $\mathcal{TC}(x)$. In addition, they propose two pruning strategies to further improve the efficiency: the *reuse technique* and the *upper bound*. The *reuse technique* avoids redundant computation in each iteration by only computing the followers in those changed components or the components of which the followers have never been computed before. The *upper bound* pruning strategy uses the upper bounds of follower numbers to reduce the number of candidate vertices that need to compute in each iteration. Specifically, for each vertex $x$, if the upper bound of the number of $x$'s followers is worse than the current optimal result, there is no need to compute $x$'s followers in the current iteration.

We can adapt GAC to solve the FM problem by replacing the coreness gain with resilience gain in *greedy anchor selection* and *upper bound pruning*. Specifically, we use an extra vertex set $V_c$ to record the vertices whose corenesses have increased before a new anchor selection. Based on Lemma 3, we know the coreness gain is the size of $F$, where $F$ is the follower set computed based on the coreness gain in GAC. Thus we replace $|F|$ with $|F \setminus V_c|$ to compute the resilience gain, and the adapted algorithm is named GAC-FM.

Although the experimental results in [35] shows that the pruning techniques highly improve the efficiency, GAC still has significant computational overheads in practice, e.g., GAC needs more than three days on the LiveJournal dataset when $b = 100$.

## A.2 More Details of Proposed Algorithms

Algorithm 5 shows how to search the solution tree by DFS. We run the paradigm by calling **GreedySearch**$(G, b, \varnothing, \varnothing, +\infty, +\infty)$. If the size of the current anchor set $A$ is larger than $b_{min}$, it is

---

**Algorithm 5**: GreedySearch($G$, $b$, $A$, $A_\neg$, $g_t$, $b_{\min}$)

**Input** : $G$ : the graph, $b$ : budget, $A$ : current anchor set, $A_\neg$ : current disregarded vertex set, $g_t$ : target resilience gain, $b_{\min}$ : current minimal budget for the target resilience gain

1 **if** $|A| \geq b_{\min}$ **then return**;
2 **if** $|A| = b$ or $g(A) \geq g_t$ **then**
3      **If** $g_t = \infty$ **then** $g_t \leftarrow g(A)$;
4      $b_{\min} \leftarrow |A|$;
5      $A' \leftarrow$**AdvGreedy**$(G, b - b_{\min})$ with $A' \cap (A \cup A_\neg) = \varnothing$;
6      Print the current solution $A' \cup A$;
7      **return**;
8 **if** $A \neq A_f$, where $A_f$ is the anchor set of the father tree node **then**
9      Compute $UB[|A|][u]$ **for** each $u \in V(G) \setminus (A \cup A_\neg)$;
10      $H[|A|] \leftarrow \varnothing$;
11 **for** each $u \in V(G) \setminus (A \cup A_\neg)$ with decreasing order $UB[|A|][u]$ **do**
12      **if** $(\cdot, u) \in H[|A|]$ **then**
13          Continue;
14      **if** $(UB[|A|][u], \cdot) > H[|A|].top()$ or $H[|A|]$ is empty **then**
15          $F \leftarrow$**FindFollowers**$(u, G, \mathcal{SC})$;
16          Push $(F, u)$ into $H[|A|]$;
17      **else Break**;
18 $(\cdot, x) \leftarrow H[|A|].top()$; $H[|A|].pop()$; $d(x) \leftarrow +\infty$;
19 $ReuseSC[|A| + 1][\cdot] \leftarrow$ **Reuse**$(x, G, A, \mathcal{SC})$;
20 **GreedySearch**$(G, b, A \cup \{x\}, A_\neg, g_t, b_{\min})$;
21 $d(x) \leftarrow |N(x)|$;
22 **GreedySearch**$(G, b, A, A_\neg \cup \{x\}, g_t, b_{\min})$;

---

pointless to continue searching the current branch hence backtrack (Line 1). Otherwise, we will backtrack in two cases: (i) $|A| = b$, we find the first result (i.e., the result of the greedy approach) (Line 2), thus set target resilience gain as the current gain (Line 3); (ii) $g(A) > g_t$, we find an anchor set for the target resilience gain (Line 2). In both cases, we need update the current minimal budget $b_{\min}$, greedily select $b - b_{\min}$ more anchors and print out the current solution, then backtrack (Lines 4-7). If the anchor set of the current tree node is different from its father's (Line 8), we compute upper bound $UB[|A|][\cdot]$ and initialize $H[|A|]$, a max heap used to store the follower results (Lines 9-10). We then compute the followers of candidate vertices sequentially in decreasing order of their upper bounds (Lines 11-17). Once get the current best anchor $x$, we remove its results from $H[|A|]$, and set its degree as infinity (Line 18). Next, we first compute the reuse results of the left child node (Line 20) and continue to search the subtree rooted at it (Line 21). We restore the anchor $x$ to a common vertex before continuing to search for the right subtree (Lines 22-23).

## A.3 Details of Compared Methods

**Shapley Value (SV).** Shapley Value is a concept in cooperative game theory. Motivated by [40], we design a Shapley Value to capture the importance of a vertex inside a vertex set. Given a vertex $v$ and a subset $A \subseteq V(G) \setminus \{v\}$, the marginal contribution of $v$ to $A$ is $g(A \cup \{v\}, G) - g(A, G)$. Let $\mathcal{P}$ be the set of all $|V(G)|!$ permutations of all the vertices in $V(G)$ and $P(v, \pi)$ be the set of vertices that appear before $v$ in a permutation $\pi$. The Shapley Value of $v$ is the average of its marginal contribution to the vertex set that appears

**Table 1: Statistics of datasets**

| Dataset | #Vertices | #Edges | $d_{max}$ | $d_{avg}$ | $k_{max}$ |
|---|---|---|---|---|---|
| **A**rxiv | 34,546 | 421,578 | 846 | 24.4 | 30 |
| **G**owalla | 196,591 | 456,830 | 14,730 | 9.2 | 51 |
| **N**otreDame | 325,729 | 1,090,108 | 10,721 | 6.5 | 155 |
| **S**tanford | 281,903 | 1,992,636 | 38,625 | 16.4 | 71 |
| **Y**outube | 1,134,890 | 2,987,624 | 28,754 | 5.3 | 51 |
| **W**iki | 557,677 | 19,197,218 | 93,188 | 51.6 | 814 |
| **L**ivejournal | 3,997,962 | 34,681,189 | 14,815 | 17.4 | 360 |
| **O**rkut | 3,072,441 | 117,185,083 | 33,313 | 76.3 | 253 |

(a) `Gowalla`  (b) `Youtube`

**Figure 8: Resilience gain v.s. Budget**

before $v$ in the permutations, i.e., $SV(v) = \frac{1}{|\mathcal{P}|} \sum_{\pi \in \mathcal{P}} g(P(v,\pi) \cup \{v\}, G) - g(P(v,\pi), G)$. Since computing the exact Shapley Value requires $\Omega(|V(G)|!)$ time, we estimate the value via sampling.
**Combinational Score (CS).** Motivated by the score function from [37], we consider the combinational effect of anchors and design a new heuristic for our problem. For each vertex $v$ in $G$ with an anchor set $A$, $\mathcal{V}^A(v) = c^A(v) + 1 - \left|\{u|u \in N(v) \wedge c^A(u) > c^A(v)\}\right|$ measures the extra supporters needed to increase $v$'s coreness by 1. Although anchoring $v$ may not increase the coreness of vertex $u$, it may provide more support for $u$, i.e., $\mathcal{V}^A(u) - \mathcal{V}^{A \cup \{v\}}(u) > 0$. Hence, we define CS considering whether the coreness of $v$ is increased or not separately, i.e., $CS(v) = score_{up}(v) + score_{nup}(v)$, where

$$score_{up}(v) = g(A \cup \{v\}, G) - g(A, G),$$

$$score_{nup}(v) = \sum_{u \in V(G) \wedge c^{A \cup \{v\}}(u) = c^{\varnothing}(u)} \frac{\mathcal{V}^A(u) - \mathcal{V}^{A \cup \{v\}}(u)}{\mathcal{V}^A(u)}.$$

We can find that for a vertex $u$, if $\mathcal{V}^A(u) - \mathcal{V}^{A \cup \{v\}}(u)$ changes after anchoring $v$, $u$ must be a neighbor of $v$ or $v$'s followers. Therefore, we can use our `AdvGreedy` to compute the followers of each candidate vertex and compute the value of CS.

## A.4 Additional Experiment Results

**Statistics of Datasets.** Table 1 shows the statistics of the datasets, ordered by the number of edges, where $d_{max}$ is the maximum vertex degree, $d_{avg}$ is the average vertex degree and $k_{max}$ is the maximum coreness of vertices in the graph.

**Exp 5: Comparison with Other Heuristics When Varying budget.** Varying budget $b$, we show the performance of all heuristics on `Gowalla` and `Youtube` in Figure 8. CS performs slightly better than `AdvGreedy` when $b \in [21, 83]$, but it needs more running time and fails when $b$ becomes larger. Results show that greedy method's advantage will become more significant as the budget increases.

**Table 2: `AdvGreedy` v.s. `Exact`**

| b | Gowalla | | | | Youtube | | | |
|---|---|---|---|---|---|---|---|---|
| | Greedy gain | exact | | ratio | Greedy gain | exact | | ratio |
| | | gain | time (s) | | | gain | time (s) | |
| 1 | 1.4 | 1.4 | 0.604 | 100% | 1.4 | 1.4 | 0.490 | 100% |
| 2 | 4.4 | 4.8 | 0.678 | 91.7% | 4.2 | 4.6 | 0.564 | 91.3% |
| 3 | 5.6 | 6.6 | 5.394 | 84.8% | 5.2 | 5.8 | 4.638 | 89.7% |
| 4 | 7.4 | 9.0 | 111.8 | 82.2% | 6.8 | 8.2 | 116.6 | 82.9% |
| 5 | 9.6 | 10.6 | 2116 | 90.6% | 8.4 | 9.4 | 2207 | 89.4% |

(a) Gowalla  (b) Youtube

**Figure 9: Performance of time-dependent framework on budget minimization problem of FM**

Besides, we can find that the rise of resilience gain of `AdvGreedy` is smooth, while others like a "staircase-style" rise, especially SV. Additionally, CS costs slightly more time than `AdvGreedy`, to compute $\mathcal{V}^A(\cdot)$ and $score_{re}(\cdot)$.

**Exp 6: Comparison with Exact Solution.** We compare `AdvGreedy` with exact algorithm which identifies the optimal $b$ anchors by enumerating all possible combinations. Due to the enormous time cost, we extract small datasets by iteratively extracting a vertex and all its neighbors, until the number of extracted vertices reaches 100. For both `Gowalla` and `Youtube`, we extract 5 subgraphs and report the average resilience gain in Table 2. The resilience gain of `AdvGreedy` is at least 82% of exact algorithm, and we find that the resilience gain ratio of `AdvGreedy` over the exact algorithm may increase with a larger budget $b$. The running time of exact algorithm is also reported in the table, while we omit that of `AdvGreedy` since it takes less than 1ms on all budgets. We can find that `AdvGreedy` is faster than the exact algorithm by up to 7 orders of magnitude.

**Exp 7: Performance on Budget Minimization Problem.** The heuristics comparisons for the budget minimization problem can also be shown in Figures 5 and 8 by swapping the x and y axis, we can find that the greedy approach obviously performs best.

Figure 9 presents the results of time-dependent framework on budget minimization problem of FM with $\lambda$ varying from 1 to 3 on `Gowalla` and `Youtube` respectively. We set the target resilience gains as the results `AdvGreedy` when $b = 100$ on both datasets. The results show that the budget continues to decrease with the running time increasing, and the minimized budgets can decrease from 100 to 90 on `Gowalla` and to 98 on `Youtube` within $10^6$ seconds.

**Exp 8: Core component tree v.s. Shell component.** We compare both the size and number of the basic units of core component tree and shell component on `Gowalla` and `Youtube`, shown in Figure 10. For vertices share the same coreness, shell component can divide them into more and smaller units compared with core component

**Table 3: Pruning techniques in GAC v.s. AdvGreedy**

| D | Component Tree | | Shell Component | | Upper Bound | |
|---|---|---|---|---|---|---|
| | $|\mathcal{T}|$ | $|\mathcal{T}_{\max}|$ | $|\mathcal{SC}|$ | $|\mathcal{SC}_{\max}|$ | $UB_\sigma > n$ | $avg\frac{UB_\sigma(\cdot)}{UB(\cdot)}$ |
| A. | 95 | 1711 | 13610 | 1711 | 62.0% | 14.86 |
| G. | 74 | 53921 | 123452 | 1862 | 10.0% | 244.0 |
| N. | 276 | 166046 | 175518 | 3035 | 0.29% | 16.58 |
| S. | 1005 | 43099 | 83797 | 23641 | 2.91% | 67.65 |
| Y. | 139 | 664726 | 873053 | 1274 | 13.9% | 929.9 |
| W. | 372 | 287809 | 434357 | 4959 | 27.6% | 94.92 |
| L. | 1755 | 818745 | 2413952 | 46965 | 27.3% | 100.6 |
| O. | 253 | 67794 | 1217084 | 41700 | 90.1% | 24.74 |

(a) Gowalla          (b) Youtube

**Figure 10: Core component tree v.s. Shell component**

tree, especially when coreness is less than 40. As Table 3 shows, the largest component size and average component size of core component tree are both much worse than shell components.

**Exp 9: Upper Bound Comparison.** We compare the upper bounds used in GAC and AdvGreedy and report the results in Table 3. A large ratio of $UB_\sigma$ in GAC exceeds $n$, e.g., 90.1% on Orkut. In the comparison of $UB_\sigma$ and our upper bound, we limit all $UB_\sigma > n$ as $n$ and compute the average value of $UB_\sigma/UB$. The results show that the average value is at least 14.86 and can reach up to 929.9.

## A.5  Proofs of Theorems

Our following analyses are all based on the theoretical results of *set cover decision* (SCD) problem [29]. The SCD problem is given a universe $U = \{u_1, \cdots, u_p\}$, a collection $\mathcal{S} = \{S_1, \cdots, S_q\}$ of subsets of $U$, and a positive integer $r$, determine if there exists a subcollection $R \subseteq \mathcal{S}$ with (i) $|R| \le r$ and (ii) $\bigcup_{S_i \in R} S_i = U$.

**Proof of Theorem 1.** Given an arbitrary instance $(U, \mathcal{S}, r)$ of the SCD problem, we build a corresponding instance of the FM problem. W.l.o.g., we assume $r < q < p$ and each $u_i$ is contained in at least one set. Figure 11 shows a construction example of 3 collections and 4 elements.

Graph $G$ contains three parts: $W$, $M$ and a $(d+1)$-clique, where $d = 2 + \max_{1 \le i \le q} |S_i|$. (a) For the $(d+1)$-clique, we arbitrarily select one vertex as the sink vertex $v_\perp$. (b) $W = \{w_1, \ldots, w_q\}$ where each $w_i$ corresponds to set $S_i \in \mathcal{S}$ in the SCD instance. (c) $M$ is a matrix with $p$ rows and $N$ columns, where $N$ is a multiple of $(p-1)$ and can be arbitrarily large. The $i$-th row in the matrix corresponds to elements $u_i \in U$ in the SCD instance. Each position of matrix $M$ contains a $d$-clique initially. For each clique in $M$, we arbitrarily select three vertices $x_{i,j}, y_{i,j}$ and $z_{i,j}$, and then modify $M$ as follows: (i) remove edges $(x_{i,j}, y_{i,j})$ and $(x_{i,j}, z_{i,j})$ from each $d$-clique; (ii) for each $i \in [1, p]$ and $j \in [1, N]$, add edges $(x_{i,j}, x_{i,j+1})$;

**Figure 11: Construction example of Lemma 1**

(iii) for each $j \in [1, N]$, add edges $(y_{i,j}, z_{f(i,j),j})$ for each $i \in [1, p]$, where $f(i, j) = ((i + ((j-1) \bmod (p-1))) \bmod p) + 1$ (making the connection between rows cycle by $p-1$); (iv) add edges from $w_k$ to $x_{i,1}$ if $u_i \in S_k$; (v) add edges from each $x_{i,N}$ to $v_\perp$ for each $i \in [1, p]$.

We can prove that the coreness of each $w_i$ is $|S_i|$, and the coreness of each vertex $v$ in $M$ is $d-2$. We then show that $G$ has the following two properties corresponding to the instance of the SCD problem:

(i) If the instance $(U, \mathcal{S}, r)$ is a *yes-instance*, then there exists an $r$-size anchor set $A$ such that $g(A, G) = r + Npd$. Consider anchoring all the $b$ vertices on $w_{i_1}, w_{i_2}, \cdots, w_{i_b}$, which are corresponding to the solution of SCD problem, then the coreness of every vertex in matrix $M$ will increase from $(d-2)$ to $(d-1)$. Let $N > (d+1+q)/(pd)$, we have $g(A, G) = r + Npd > Npd > Npd/2 + (d+1+q)/2 = (Npd + d + 1 + q)/2 = n/2$. Therefore, $g(A, G) = \Omega(n)$.

(ii) If $(U, \mathcal{S}, r)$ is a *no-instance*, then there exists at least an $i$-th row in $M$, in which the corenesses of all the vertices will not increase. Therefore, these vertices will be removed in core decomposition when $k = d - 1$. Note that $N$ is a multiple of $(p-1)$, for each row in $M$ we denote positions $(i-1) * (p-1) + 1$ to $i * (p-1)$ by $patch_i$. Then for each $j$-th row where $j \ne i$, there exists at least one vertex in each $patch_i$ which is adjacent to a vertex in the $i$-th row, i.e., if no anchor is placed in each $patch_i$, this patch will also be removed when $k = d - 1$ via the core decomposition. Thus $r$ anchors can obtain at most $r(p-1)d + r$ resilience gain. Since $d > 2$, we have $r(p-1)d + r < rpd$. As $r$ is corresponding with budget $b$ in the FM instance, i.e., $r = b$, we can ensure that $g(A, G) = O(b)$. □

**Proof of Corollary 1.** According to Lemma 1, for each instance $(U, \mathcal{S}, r)$ of the SCD problem, it is a *yes-instance* iff there is a $r$-size anchor set s.t. the resilience gain is $\ge r + Npd$ in the corresponding FM instance. If there is a polynomial-time solution for the FM problem, then we can determine in PTIME whether the optimal resilience gain exceeds $r + Npd$, and subsequently solve the SCD problem in PTIME. □

**Proof of Theorem 3.** We prove this theorem by an FPT-reduction from the well-known W[2]-hard SCD problem parameterized by the size of set cover [8]. Consider an arbitrary instance $(U, \mathcal{S}, r)$ of the SCD problem, we construct a corresponding instance of the FM problem on a graph $G$. For each $S_i \in \mathcal{S}$, we create a vertex $w_i$ in $G$. For each $u_i \in U$, we create a vertex $m_i$ with $p$ cliques connected to it, where each clique is a $(p+2)$-clique. Finally, we add edges between $m_i$ and $w_j$ if $u_i \in S_j$. Note that the budget $b$ in

the FM instance corresponds to the size $r$ of the set cover in the SCD instance.

We next prove that the SCD instance $(U, \mathcal{S}, r)$ is a *yes-instance* iff there exists an anchor set $A \subseteq V(G)$ with $|A| \leq b$ that the corresponding resilience gain $g(A, G) \geq b + p$.

In one direction, we assume that the SCD instance is a *yes-instance*. In graph $G$, we know that the coreness of each $w_i$ is $|S_i|$, and the coreness of each $m_i$ is $p$. According to the solution $\{S_{i_1}, \cdots, S_{i_r}\}$ of the SCD instance, we can anchor the corresponding vertices $w_{i_1}, \cdots, w_{i_r}$ in $G$, thus the resilience gain is $b + p$. Hence we can conclude that the resilience gain is at least $b + p$.

For the other direction, we prove by contradiction, i.e., assume that the SCD instance is a *no-instance*. Given that in $G$, anchoring each $m_i$ or vertex in cliques can obtain only 1 resilience gain, we consider placing anchors in $w_i$, which can get extra gain in $m_j$ if edge $(w_i, m_j)$ exists. As there exists no set cover of size $r$, we can obtain at most $b + p - 1$ resilience gain after anchoring $A$ with $|A| < b$, when we place $b$ anchors on $w_{i_1}, w_{i_2}, \cdots, w_{i_b}$. Hence there exists a contradiction to that there exists an anchor set $A$ with $|A| \leq b$ that $g(A, G) \geq b + p$. □

**Proof of Theorem 4.** Suppose there exists an anchor vertex set $A$ and a vertex $x \notin A$, anchoring new vertex $x$ cannot decrease other vertices' corenesses. Thus we have $g(A) \leq g(A \cup \{x\})$, which means the function $g(\cdot)$ is monotonic.

If $g(\cdot)$ is submodular, for two arbitrary vertex set $A$ and $B$, it must hold that $g(A) + g(B) \geq g(A \cup B) + g(A \cap B)$. Consider a graph $G$ with a vertex set $V = \bigcup_{1 \leq i \leq 5} v_i$, the vertices in $\bigcup_{1 \leq i \leq 3} v_i$ form a 3-clique, $v_4$ connects to $v_1$ and $v_2$, and $v_5$ connects to $v_3$. If $A = \{v_4\}$ and $B = \{v_5\}$, $g(A) + g(B) = 2 < g(A \cup B) + g(A \cap B) = 5$, thus $g(\cdot)$ is non-submodular. □

**Proof of Theorem 5.** From Lemma 2 and the definition of upstair path, if a vertex $u$ is a follower of anchor $x$, it must be in set $SN(x)$ or connected to a vertex $v \in SN(x)$ through a path where each vertex has the same coreness $c(v)$. By the definition of shell component, $u$ and $v$ are in the same shell component, thus shell components in $CS(x)$ contain all the followers of vertex $x$. □

**Proof of Theorem 6.** Recall the analysis in Section 5.2, $F[v][S]$ will not change after anchoring $x$ if the supporters of vertices in $S$ do not change. As $S \in ReuseSC(v)$ does not contain any vertex in $V'^*$, $S$ will remain the same and the anchor $x$ is not a supporter of any vertex in $S$. Besides, we consider the supporters of vertices in $S$ by considering the shell components $S'$ with at least one edge between $S'$ and $S$: (i) $S'.c > S.c$, as $S'.c$ will not decrease, the vertices in $S'$ who are supporters of vertices in $S$ before anchoring $x$ will still be supporters of them after anchoring $x$; (ii) $S'.c < S.c$, since $S$ does not change after anchoring $x$, $S'.c$ is at most $S.c - 1$, thus the vertices in $S'$ are still not the supporters of vertices in $S$; (iii) no $S'$ with $S'.c = S.c$, otherwise $S'$ and $S$ are the same component. □

**Proof of Theorem 7.** By Lemma 2, we know that a follower of vertex $x$ must be included in its upstair DAG. In a shell component $S$, every vertex in $x$'s upstair DAG is counted at least once in $UB(x, S) = \min\left\{|S.V \cap U(x)|, \sum_{u \in SN(x) \cap S.V} UB(u, S)\right\}$. Therefore we have $|F[x][S]| \leq UB(x, S)$, thus $g(A \cup \{x\}, G) - g(A) = \sum_{S \in CS(x)} F[x][S] \leq \sum_{S \in CS(x)} UB(x, S) = UB(x)$. □

