# OpenReview forum: "Optimizing Network Resilience via Vertex Anchoring"
_ACM.org/TheWebConf/2024/Conference — TheWebConf24 Oral_

### Official Review · Reviewer_yXme · 2023-11-21

**Novelty:** 2
**Technical Quality:** 4

**Review:**

The paper proposes an algorithm to increase the core resilience of the graph. Core resilience is defined as the number of nodes whose core number increases when a new set of anchors are given.

- The major issue with the paper is the motivation for the objective function (Def 4, resilience gain), which is simply the #nodes whose core number increases. There are some superficial arguments in intro but no empirical or theoretical justification is provided for why such an objective is better at measuring core resilience. Indeed, [35] has a very similar function: they consider sum of core number increases across all nodes (instead of #nodes whose core number increases). The algorithms and theoretical findings are quite similar to [35].
- In experiments, where is GAC-FM in Fig 5? It's probably the most obvious comparison that needs to be made but omitted for some reason (it's included for runtime comparison in Fig 6a).

**Questions:**

See above.

**Reviewer Confidence:**

4: The reviewer is certain that the evaluation is correct and very familiar with the relevant literature

**Scope:**

2: The connection to the Web is incidental, e.g., use of Web data or API

---

### Official Review · Reviewer_Ynxq · 2023-11-24

**Novelty:** 4
**Technical Quality:** 5

**Review:**

-Summary

This paper focuses on enhancing network resilience, defined as a network's ability to maintain functionality. It introduces the follower maximization problem, aimed at increasing the coreness of network vertices within a budget, a task proven to be NP-hard and W[2]-hard. The approach involves an advanced greedy algorithm and a time-dependent framework to efficiently find solutions. The paper also highlights the bias in engagement enhancement in existing methods and the need to consider the engagement dynamics of all users. The effectiveness of these methods is validated through experiments on real-life datasets, contributing significantly to the field of network resilience.

-Strong points

S1. This paper introduces the novel follower maximization problem in network resilience. This approach focuses on enhancing the coreness of network vertices within a budget constraint, addressing a critical aspect of network theory that has not been extensively explored before.

S2. This paper’s findings are supported by experiments using real-life datasets.

S3. This paper is well-written and easy to understand.

-Weak points

W1. The overall framework of this paper seems very similar to a previous research [35]. For example, previous research [35] uses FindFollowers based on CoreDecomp and BuildCCT and employs ResultReuse to improve its efficiency. This paper leverages FindFollowers based on ShellDecomp and adopts Reuse to save time. The similarity may reduce the overall novelty. More justification on novelty should be given.

W2. AdvGreedy employs a greedy strategy based on Follower Computation on Shell Component. My question is, why do we only search for followers within the Shell Component, instead of using the Upstair Path to look for followers in a deeper layer of the shell? This approach might enable finding all followers in one go. Are there any specific considerations behind this approach?

W3. In Algorithm 4, 'g' (the set of vertices whose coreness has changed) seems to be updated in each iteration, but I could not find the updating process within the algorithm.

W4. While the time complexities of the components of the main algorithm AdvGreedy (algorithms 1, 2, and 3) are analyzed, the time complexity of AdvGreedy itself has not been analyzed.

W5. Experiments were conducted with a large budget variation ({1, 100, ..., 500}). I am curious whether AdvGreedy maintains its advantage over other baselines with a reduced, small budget variation ({1, 10, ..., 50}).

W6. Section 5.4 states that this paper identifies a tighter upper bound than previous research. I am curious about how much better this new tighter upper bound is compared to the previous one in experiment.

W7. This paper lacks comparisons with other previous research [31, 35, 52], as its baselines are solely heuristic methods proposed within the study itself.  Additionally, I am interested in how its performance compares with methods capable of finding optimal solutions in the time-dependent framework, such as ILP and branch-and-bound.

W8. Minor issue: At the beginning of section 5.2, Lemma 3 is mentioned, but it is not easily found. It would be helpful to include Lemma 3 in the main text.

W9. Minor issue: In section 5.1, it is suggested to provide a formal definition of 'layer' (deletion sequence of core decomposition) for clarity.

W10. Minor issue: In Algorithm 4, the set of anchored vertices A should be initialized to an empty set at the beginning.

W11. Minor issue: It is helpful to do ablation study of each pruning strategy of the time-dependent framework.


---
# Comments after Rebuttal
I do appreciate the authors' response. However, I believe that due to the space and time constraints, the authors cannot fully revise the paper to address all my concerns. Therefore, I am keeping my rating on this paper.

**Questions:**

W1-W7

**Reviewer Confidence:**

3: The reviewer is confident but not certain that the evaluation is correct

**Scope:**

3: The work is somewhat relevant to the Web and to the track, and is of narrow interest to a sub-community

---

### Official Review · Reviewer_7pKC · 2023-11-24

**Novelty:** 2
**Technical Quality:** 5

**Review:**

Positives:

- It is good to see the classical core resilience problem via vertex anchoring.

- The greedy algorithm is effective and the paper has done a good job to make it work in practice.

- The experiments include many results.

Negatives:

I honestly think the work is well-executed. However, I am a little concerned about novelty given the work in this space ("...maximizing 𝑘-core via vertex anchoring [6, 13, 31, 37, 52], minimizing the size of 𝑘-core via
vertex removal [53, 57], and edge manipulations [25, 40, 58, 59]" and [17].) Usually these works show similar hardness results and then use some versions of greedy (or even better algorithms such as SV). So, a paragraph explaining the novelty would help.

- The theoretical results are known for a small variation of the problem. So it is not really surprising.

- Some baselines (e.g., SV) are not well-explained. The results are a little unexpected.

- Some results are observations and unnecessarily has shown as Theorems (e.g., Theorem 6).

**Questions:**

1. After proving Lemma 1, does Theorem 4 add any value?

2. The point of Theorem 4 is usually to show if Greedy might have approximation. However, you use Greedy to solve your problem. So, the motivation is a little unclear.

3. How is SV implemented? How do you design your final solution from SV? For instance, do you select the top ones? Or, do you select them adaptively? Usually, SV captures higher combinatorial properties than usual greedy. Could you please discuss it?

**Reviewer Confidence:**

4: The reviewer is certain that the evaluation is correct and very familiar with the relevant literature

**Scope:**

3: The work is somewhat relevant to the Web and to the track, and is of narrow interest to a sub-community

---

### Official Review · Reviewer_r1Pp · 2023-11-26

**Novelty:** 5
**Technical Quality:** 6

**Review:**

This paper aims to find the coverage of followers of given user sets. By considering the follower coverage problem as a follower maximization problem which is another kind of influence maximization problem. The follower maximization problem is proven to be a np-hard problem and a greedy method is proposed to solve it. The number of followers is named as resilience gain. This paper proposes proper example to show the node engagement and coreness on certain user to show the motivation of the FM problem.

Detail comments
1.    The basic theory is familiar with influence maximization problem, however, the discussion about why we don’t use influence maximization method to solve the FM problem is not clearly proposed. More discussions are needed.
2.  Traditional influence maximization method should be compared.

**Questions:**

Why donot you use influence maximization method to solve the FM problem?

**Reviewer Confidence:**

4: The reviewer is certain that the evaluation is correct and very familiar with the relevant literature

**Scope:**

4: The work is relevant to the Web and to the track, and is of broad interest to the community

---

### Decision · Program_Chairs · 2024-01-22

**Decision:**

Accept (Oral)

**Comment:**

The reviewers appreciated the formulation of the follower maximization problem, hardness analysis, and the design of the greedy heuristic. In general, the reviewers found the paper well-written and the technical quality adequate. In contrast, the reviewers had serious reservations about the novelty of the algorithms vis-a-vis the large body of related work in problems such as influence maximization. Indeed, there was an unusually large difference in the reviewers' scores for technical quality and novelty. Overall, the paper is borderline.